# Antimicrobial Resistance Patterns of *Staphylococcus aureus* Cultured from the Healthy Horses’ Nostrils Sampled in Distant Regions of Brazil

**DOI:** 10.3390/antibiotics14070693

**Published:** 2025-07-09

**Authors:** Mauro M. S. Saraiva, Heitor Leocádio de Souza Rodrigues, Valdinete Pereira Benevides, Candice Maria Cardoso Gomes de Leon, Silvana C. L. Santos, Danilo T. Stipp, Patricia E. N. Givisiez, Rafael F. C. Vieira, Celso J. B. Oliveira

**Affiliations:** 1Department of Animal Science, College for Agricultural Sciences, Federal University of Paraiba (CCA/UFPB), Areia 58397-000, PB, Brazil; saraiva_ufba@hotmail.com (M.M.S.S.); candice.zoo@hotmail.com (C.M.C.G.d.L.); silvanalimazoo@hotmail.com (S.C.L.S.); patriciagivisiez@gmail.com (P.E.N.G.); 2School of Agricultural and Veterinary Sciences (FCAV/Unesp), São Paulo State University, Jaboticabal 14885-900, SP, Brazil; heitor.leocadio@unesp.br (H.L.d.S.R.); valpbenevides@gmail.com (V.P.B.); 3College for Natural Sciences, Federal University of São Carlos, São Carlos 13565-905, SP, Brazil; danilostipp@gmail.com; 4Department of Epidemiology and Community Health, The University of North Carolina at Charlotte, Charlotte, NC 28223, USA; 5Center for Computational Intelligence to Predict Health and Environmental Risks (CIPHER), The University of North Carolina at Charlotte, Charlotte, NC 28223, USA

**Keywords:** bacterial carriage, *blaZ*, equine, MRSA, multidrug resistance

## Abstract

*Staphylococcus aureus* (*S. aureus*) is a major cause of opportunistic infections in humans and animals, leading to severe systemic diseases. The rise of MDR strains associated with animal carriage poses significant health challenges, underscoring the need to investigate animal-derived *S. aureus*. Objectives: This study examined the genotypic relatedness and phenotypic profiles of antimicrobial resistance in *S*. *aureus*, previously sampled from nostril swabs of healthy horses from two geographically distant Brazilian states (Northeast and South), separated by over 3700 km. The study also sought to confirm the presence of methicillin-resistant (MRSA) and borderline oxacillin-resistant (BORSA) strains and to characterize the isolates through molecular typing using PCR. Methods: Among 123 screened staphylococci, 21 isolates were confirmed as *S. aureus* via biochemical tests and PCR targeting species-specific genes (*femA*, *nuc*, *coa*). Results: REP-PCR analysis generated genotypic profiles, revealing four antimicrobial resistance patterns, with MDR observed in ten isolates. Six isolates exhibited cefoxitin resistance, suggesting methicillin resistance, despite the absence of the *mecA* gene. REP-PCR demonstrated high discriminatory power, grouping the isolates into five major clusters. Conclusions: The genotyping indicated no clustering by geographical origin, highlighting significant genetic diversity among *S. aureus* strains colonizing horses’ nostrils in Brazil. These findings highlight the widespread and varied nature of *S. aureus* among horses, contributing to a deeper understanding of its epidemiology and resistance profiles in animals across diverse regions. Ultimately, this genetic diversity can pose a public health risk that the epidemiological surveillance services must investigate.

## 1. Introduction

*Staphylococcus* spp. comprises a group of mesophilic bacteria commonly found in the environment and colonizing the skin and membranes of humans and animals [1,2]. *Staphylococcus aureus* (*S. aureus*) is typically a coagulase-positive staphylococci (CoPS) and is considered the main causative agent of opportunistic staphylococcal infections in both humans and animals, ranging from localized self-limiting cutaneous lesions to severe systemic infections [2,3].

A major concern regarding staphylococcal infections is the emergence of multidrug-resistant (MDR) strains, particularly methicillin-resistant *Staphylococcus aureus* (MRSA) [1]. This emergence poses a significant threat to public health in all sectors, reducing the availability of antibiotics used in hospitals and making the treatment of bacterial infections more challenging [4]. In another aspect, antimicrobial-resistant bacteria have a significant economic impact in low-income countries, increasing the health burden and mortality among vulnerable populations, which could reach ten million annual deaths and cost a hundred trillion dollars by 2050 [5].

Reports have shown that farm and pet animals can be asymptomatic carriers of MRSA [3,6], as well as borderline oxacillin-resistant *Staphylococcus aureus* (BORSA) [7,8]. In this scenario, resistant *S*. *aureus* can spread depending on the movement of the animals and even in human–animal interactions, creating a risk of infection to the farmers, workers, and tutors even when in contact with healthy animals [9,10]; moreover, MRSA has been described as having a high potential for transfer between equine species [11]. On the other hand, *Staphylococcus* spp. can also be carried and spread by asymptomatic humans, completing the epidemiologic infection chain of this pathogen and making it a challenge to control [1,9]. Additionally, genetic approaches have identified these bacteria as the causative agents of infections in humans who come into contact with livestock, and they may represent a significant risk to public health [6,10].

Methicillin-sensitive *S. aureus* (MSSA), MRSA, and BORSA are often detected in the nostrils of healthy horses in developed countries. For instance, Pusterla et al. [12] presented the detection of MRSA in 22%, while Hryniewicz and Garbacz [13] reported a prevalence of more than 25% of BORSA in equines attended to by routine veterinary services; conversely, Mama et al. [14] found 90% staphylococci prevalence in horses intended for human consumption. Besides β-lactams, resistance to aminoglycosides, fluoroquinolones, folate pathway inhibitors, fusidic acid, macrolides, and tetracyclines has been reported in *S*. *aureus* from equines in recent years, in different parts of the world [11,15,16]. It is important to emphasize reports of healthy horses carrying heavy-metal-resistant *S*. *aureus* [16] and a high nasal colonization rate by MDR strains, which can reach one in every four equines [11].

However, unlike the existing data for other livestock species, there is a lack of information regarding the presence of *S*. *aureus* in low-income countries. In a study conducted in Brazil, a clonal similarity of *S. aureus* isolated from the nostrils of hospitalized horses and veterinary hospital staff was observed, reinforcing the need for surveillance measures to monitor MDR staphylococci [17] that can cause infections in both humans and animals. In this context, this study aimed to investigate the antimicrobial resistance patterns and genetic relatedness of *Staphylococcus aureus* isolates from healthy horses in two distant regions of Brazil. Key investigations include the confirmation of MRSA and BORSA strains and the identification of clonal relationships through molecular typing, contributing to the understanding of the genetic diversity of resistant *S*. *aureus* in horse populations.

## 2. Results

Out of the 123 isolates studied, 90 (73.17%) were confirmed as CoPS isolates via the tube-coagulase test, but only 21 were shown to be positive for at least one of the three *S. aureus* species-specific genes. The other isolates neither produced free coagulase nor were they positive for the *S. aureus* markers; therefore, they were identified as coagulase-negative staphylococci (CoNS).

In the molecular identification, the genes *nuc* and *femA* were identified in 16 isolates, whereas only 9 isolates were positive for the *coa* gene. As shown in Table 1, the concordance amongst the presence of these genes used to confirm *S. aureus* was low to fair. However, one of these genes was found exclusively in all *S*. aureus isolates, confirming the specificity of these genes in *Staphylococcus* species.

The data in Table 2 present the frequency patterns observed in the present study, where the most common patterns were *nuc-femA* and *nuc-femA-coa*, shared by 33% and 28.5% of *S. aureus* strains, respectively. The gene *coa* was the least frequent gene found in *S. aureus*.

According to the Kirby–Bauer antimicrobial susceptibility test, higher rates of resistance were observed in the studied strains for ampicillin (47.6%; 10/21), penicillin G (47.6%; 10/21), tetracycline (38.10%; 8/21), and vancomycin (38.10%; 8/21). On the other hand, the lowest resistance rates were observed for amoxicillin-clavulanate and enrofloxacin (4.76%; 1/21). The other AMR rates found were clindamycin (33.3%; 7/21), cefoxitin (28.57%; 6/21), oxacillin (23.8%; 5/21), chloramphenicol (14.28%; 3/21), azithromycin (9.52%; 2/21), and gentamicin (9.525%; 2/21). Pan-susceptibility was observed in six (28.57%) isolates.

A total of four different antimicrobial-resistant patterns were observed, and ten (47.6%) MDR isolates were identified. Other results included six (28.6%) pan-susceptible isolates, three (14.3%) resistant to two antimicrobials, and two (9.5%) resistant to only one antimicrobial.

Regarding the genotype pattern of antimicrobial resistance, a total of 9 out of 123 *Staphylococcus* isolates (7.3%) were positive for the *mecA* gene. However, this gene was not found among the 21 *S. aureus* isolates, including the 6 isolates that were resistant to cefoxitin, indicating that these isolates were likely MRSA. No *mecC* was found in any of the 123 *Staphylococcus* isolates investigated in this study.

Unlike *mec* genes, *blaZ,* a penicillinase determinant, was found in 27 of 123 *Staphylococcus* spp., including 5 *S*. *aureus*. Interestingly, all *S*. *aureus* strains harboring the *blaZ* gene were phenotypically resistant to ampicillin, cefoxitin, oxacillin, and penicillin; three of them also accumulated vancomycin resistance (Table 3).

Concerning the genotyping of *S*. *aureus* from this study, large genetic diversity was observed (Figure 1). The Rep-PCR using the primer RW3A showed a D-value of 0.96, indicating that it is highly discriminatory. Similar results were described by Silva et al. [18] and Leon et al. [19]. The *S. aureus* isolates were clustered in five major groups (A, B, C, D, and E). Cluster A comprised two *S. aureus* isolates from State B and the MRSA USA400. Except for cluster A, which indicates the presence of MRSA among the *S. aureus* investigated in the present study, the isolates were not clustered according to their geographical origin, indicating a large genotypic diversity of *S. aureus* colonizing the nasal cavities of horses in Brazil. Isolates from different geographical regions were identified in all clusters in the present study.

## 3. Discussion

The low occurrence of *S*. *aureus* amongst staphylococci might be related to the fact that only healthy animals were sampled. Similar frequencies of *S. aureus* carriage in horses have been reported in Switzerland, Libya, and Brazil [20,21,22].

The unexpected frequency of the *coa* gene amongst isolates, even in coagulase-producing bacteria, could be related to the putative polymorphism of this gene. Variations in the *coa* gene have been previously observed amongst *S. aureus* [20,23]. Importantly, although PCR targeting the *nuc* gene has been typically used for *S. aureus* confirmation purposes due to its specificity for this bacterial species [24,25], our findings suggest that the use of more markers might contribute to improving *S. aureus* detection in strains exclusively sampling from horses.

Nosocomial infections in horses have been attributed to MDR *S. aureus,* and the high resistance rates against drugs commonly used are of great concern because of the therapeutic implications associated with multi-resistant strains in horses [14,20,22]. Furthermore, a worrying finding of our study was the coexistence of both β-lactam and vancomycin resistance. Concerning these two classes of antimicrobials, the β-lactam drugs have been indicated by the World Health Organization (WHO) as being critically important for human health [26], and vancomycin is the chosen drug for treating infections caused by methicillin-resistant staphylococci [27].

In this study, we found eight *S. aureus* strains resistant to ß-lactam and eight resistant to vancomycin (five of them were resistant to both). Although a limitation of the study may be the antimicrobial susceptibility test by Kirby–Bauer to identify vancomycin-resistant strains, previous reports using disk diffusion have established a high correlation between vancomycin-resistant *S*. *aureus* and both minimum inhibitory concentration (MIC ≥ 16 µg/mL) and the presence of genetic determinants (*vanA* and *vanB*) [28,29].

The correct identification of *S. aureus* strains is not a straightforward task, and it is believed to introduce a significant bias in studies aiming to assess antimicrobial resistance in *S. aureus*, especially in MRSA strains [8,25,30]. The low occurrence of the *mec* gene is not uncommon. In a previous study conducted in Brazil, only 1% of the *Staphylococcus* spp. isolated from horses presented the *mecA* gene, all of them identified in *S. pseudintermedius* [17]. Recent studies have demonstrated that the MRSA phenotype does not necessarily require the presence and expression of *mecA*, and other putative mechanisms for methicillin resistance, such as *mecC*, as well as hyperexpression of the *bla*-operon, have also been shown to occur in methicillin/cefoxitin-resistant *S*. *aureus* [13,31,32]. This is particularly important for animal-derived *S. aureus*, since the *mecA* gene was initially targeted as a marker for methicillin resistance in *S. aureus* from humans [33]. In this sense, pleomorphism in *mec*-operon might occur, making it challenging to confirm MRSA using PCR-*mecA* [34]; however, this hypothesis is based solely on previously published data, as no sequencing was performed in this study, which can be seen as a limitation of the study. Notably, the six cefoxitin-resistant isolates identified in the present study were also MDR isolates (Table 3), which have invariably been associated with MDR [33].

The potential of *blaZ*-positive *S*. *aureus* to show resistance to various β-lactam drugs leads to the misidentification of BORSA, MRSA, or MSSA. This condition implies the treatment failure of BORSA-associated infections when the bacterial identification is based on cefoxitin susceptibility [26]. Therefore, the correct identification of BORSA is crucial for the success of drug therapy in human medicine, as well as for the surveillance of bacteria in horse farms, focusing on its role in One Health [27].

Staphylococcosis in animals has been linked to pre-existing conditions associated with chronic diseases [35], and can cause serious diseases, especially when strains carry resistance determinants. MRSA infection caused by a strain carrying *mec* and *blaZ* genes can result in chronic obstructive pulmonary disease (COPD) and thrombophlebitis [31]. In that case, veterinarians and treaters are affected and need to be treated against *S*. *aureus* infection [31].

Although BORSA infections in humans are infrequently identified, a recent case report has been published, resulting in one death from a nosocomial infection, and observations of high genetic heterogeneity among *S*. *aureus* isolates, as well as the identification of *blaZ* gene expression. [36]. A more severe outbreak was reported by Huang et al. [37], with a high human mortality rate, reaching 24.6%, caused by BORSA strains in affected patients of bacteremia. Conversely, recent reports have found a high prevalence of MDR BORSA strains in healthy horses [38], which can serve as a source of infection for farmers and handlers.

BORSA strains have been reported in other animal species, such as asymptomatic swine [8] and in the nasal cavities of patients hospitalized in a neonatal intensive care unit [26]. Curiously, from both swine and human cases, sequence type 398, related to the LA-MRSA lineage, was reported, despite the absence of *mecA*, *mecB*, or *mecC* genes [8,26]. These previous reports and our results on the occurrence of *S*. *aureus* strains in humans and healthy animals with methicillin-resistant phenotypes highlight its potential impact on public health.

Although the occurrence of *S*. *aureus* in equines presenting no clinical signs is not uncommon, lineages originally from animals have been found more often in humans. For instance, the prevalence of ST398 could be 10-fold higher in the nasal cavities of humans than in horses [17], corroborating our findings and confirming that equines and humans are potential reservoirs of this pathogen.

Unlike our study, Santos et al. [8] identified a lower diversity of clusters in *S*. *aureus* isolates from swine, which is consistent with their study in a similar geographical region, suggesting that the reservoir for this bacterium may be more closely linked to the animal species than its origin. However, the genetic similarity of MRSA USA 400 with BORSA from swine [8], as found in horses, poses a potential health risk to handlers, animals, and individuals involved in management [39].

Although the study regions are approximately 3700 km apart, genetic compatibility was observed between clusters B and C of some *S*. *aureus* isolates (Figure 2). The identification of these genetically similar isolates from animals in different regions may indicate the dissemination of *S*. *aureus*-resistant strains in other states of the country, hosted by both humans and equines, due to their strong interaction [3]. However, our results are somewhat divergent from those obtained from other animal species. Recent studies focusing on *S*. *aureus* frequency from Brazil highlighted a large diversity of *S*. *aureus* strains from swine, carrying resistance determinants [8,40]. Therefore, we hypothesize that two factors could be involved in the dissemination of *S. aureus* in a continental Country: the selective pressure imposed by off-label antimicrobials used in farm animals [4], and the transport of animals and humans between different geographical regions, carrying MDR *S*. *aureus* strains [41]. Ultimately, although the lack of clustering by region may be unexpected, these results can help veterinarians and personnel handling horses in the prudent use of antimicrobials in veterinary medicine [3].

## 4. Materials and Methods

### 4.1. Bacteria and Staphylococcus aureus Confirmation

Samples of nasal swabs from clinically healthy horses were previously and randomly collected from animals originating from different Brazilian states: seven cities in Rio Grande do Sul State (identified as AI to AVII) and one municipality in Paraiba State (identified as B). The approximate distance between them is 3700 km (Figure 2).

A total of 123 *Staphylococcus* spp. strains were received in the Laboratory for Animal-derived Foods (LAPOA) of the College for Agricultural Sciences, Federal University of Paraíba (CCA/UFPB): 87 isolates from State A, and 36 isolates from State B. *S*. *aureus* identification was performed using conventional morphological (stained Gram) and phenotypic (biochemical tests, including coagulase production, Voges–Proskauer, polymyxin B resistance, and mannitol fermentation) analysis. Isolates showing morphological and phenotypical *S. aureus* characteristics were confirmed by polymerase chain reaction (PCR) targeting three specific genes (*nuc*, *femA*, and *coa*), according to the work of Saraiva et al. [24] (Appendix A Table A1).

The extraction of genomic DNA from *Staphylococcus* spp. was conducted using the boiling-centrifugation method [42]. Briefly, five colonies of each isolate were suspended in 100 μL of ultrapure water, frozen for 10 min, and then boiled at 100 °C for an additional 10 min. The samples were centrifuged for 3 min at 12,000 rpm at 4 °C to remove any cellular debris. The supernatant of each sample was carefully removed, and, to avoid contaminants, 70 μL was transferred to another tube and then stored at 4 °C until use. After DNA extraction, the master mix was prepared in a 25 μL volume using 1 μL of Taq DNA polymerase (LGC Biotechnologies, Middlesex, England), two μL of MgCl2, 20 μM of each dNTP (Thermo Fisher Scientific, Agawam, MA, USA), and two μL of DNA template [24]. PCRs were performed in TC5000 thermal cycler (Techne, Essex, England); amplification products were electrophorized in 2% agarose gel, stained with Gelred (Biotium, Fremont, CA, USA), visualized under UV, and documented by Gel Logic 212 PRO (Carestream Molecular Imaging Software—Version 5.0, Carestream Health Inc., Rochester, NY, USA).

### 4.2. Antimicrobial Susceptibility Test

An antimicrobial susceptibility test was performed using the Kirby–Bauer disk-diffusion method, as described by the Clinical and Laboratory Standards Institute [43]. The antimicrobial drugs used and their respective concentrations were as follows: amoxicillin/clavulanate (AMC, 10 µg), ampicillin (AMP, 30 µg), azithromycin (AZI, 15 µg), cefoxitin (CFO, 30 µg), chloramphenicol (CLO, 30 µg), clindamycin (CLI, 2 μg), enrofloxacin (ENO, 10 µg), gentamicin (GEN, 10 µg), oxacillin (OXA, one µg), penicillin G (PEN, 10 UI), tetracycline (TET, 30 µg) and vancomycin (VAN, 30 µg) (CECON^®^, São Paulo, Brazil). The interpretation of antimicrobial susceptibility test results was performed according to the CLSI [43]. Isolates were considered MDR when showing resistance against three or more drugs derived from different antimicrobial classes.

### 4.3. MRSA and BORSA Confirmation, and Genotyping

Confirmed *S. aureus* isolates were extracted via the technique of phenol: chloroform: isoamyl alcohol, according to the Laboratory Manual of Sambrook et al. [44]; afterwards, investigation by PCR targeting *mecA*, *mecC* [8], and *blaZ* [45] genes was performed, using the same master mix above (Appendix A Table A1). Two strains (USA 400 and ATCC 25923) were used as positive and negative controls. Amplification products were electrophorized in 2% agarose gel, stained with Gelred (Biotium, Fremont, CA, USA), visualized under UV, and documented by Gel Logic 212 PRO (Carestream Molecular Imaging Software—Version 5.0, Carestream Health Inc., Rochester, NY, USA). Additionally, these were genotyped by Repetitive Extragenic Palindromic-PCR (Rep-PCR), targeting the RW3A, according to the work of Zee et al. [46]. All PCRs were performed into TC5000 thermal cycler (Techne, Essex, England), and amplification products were electrophorized in 2% agarose gel, stained with Gelred (Biotium, Fremont, CA, USA) visualized under ultraviolet light, and documented by Gel Logic 212 PRO (Carestream Molecular Imaging Software—Version 5.0, Carestream Health Inc., Rochester, NY, USA).

### 4.4. Data Analyses

The concordance amongst the presence of the genes *nuc*, *femA,* and *coa* was determined by the Cohen’s kappa index of concordance according to the criteria previously established by Landis and Koch [47] (Appendix A Table A2).

The genetic relatedness among *S. aureus* was determined by analyzing the similarity among Rep-PCR fingerprints using the Dice coefficient with a 2% tolerance for genetic distance measurement calculations. Cluster analysis was performed using the unweighted pair-group method with average linkages (UPGMA) via commercial software (BioNumerics 7.1, Applied Maths, Sint-Martens-Latem, Belgium), and the results were presented in a dendrogram. An 80% similarity coefficient was used as a threshold for clustering. The discriminatory power (D value) was calculated as described by Hunter [48].

## 5. Conclusions

In conclusion, our findings indicate a high genetic diversity of *S. aureus* colonizing the nasal cavities of horses from Brazil, as well as the presence of BORSA among the samples. A special concern is the high frequency of MDR pathogens harbored by horses, which could affect not only animals but also human health. Our results demonstrate that the MRSA phenotype does not require the presence or expression of the *mecA* gene. Based on the literature, this indicates pleomorphism in the presence of the *mec*-operon and *bla*-operon, making it difficult to confirm whether a strain is carrying the gene.

## Figures and Tables

**Figure 1 antibiotics-14-00693-f001:**
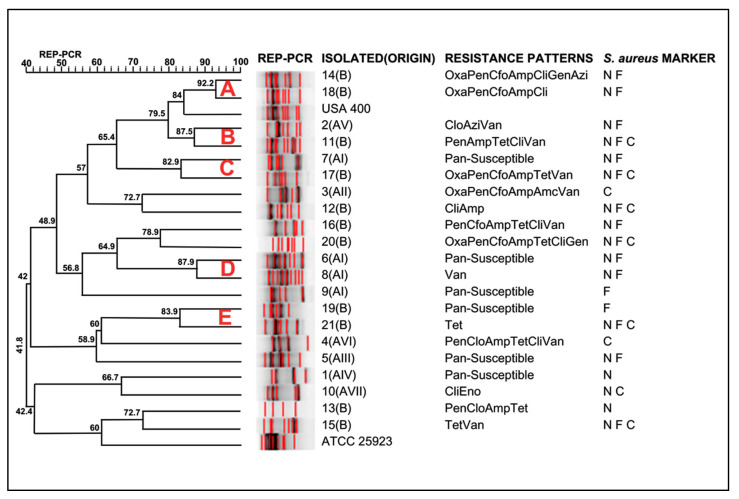
Dendrogram of genotypic analysis made for RW3A from 21 *S. aureus* isolates. Amp = ampicillin; Cfo = cefoxitin; Pen = penicillin G; Clo = chloramphenicol; Gen = gentamicin; Oxa = oxacillin; Cli = clindamycin; Tet = tetracycline; Amc = amoxicillin/clavulanate; Azi = azithromycin; Van = vancomycin; Eno = enrofloxacin; N = Positive for *nuc* Marker; F = Positive for *femA* Marker; C = Positive for *coa* Marker.

**Figure 2 antibiotics-14-00693-f002:**
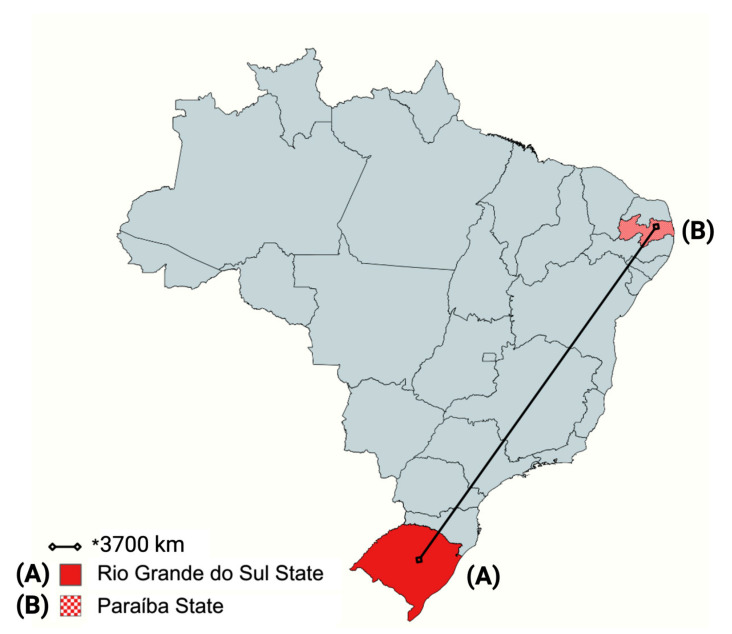
Distance between the State of Rio Grande do Sul and the State of Paraíba in a map of Brazil. The bacteria were isolated from the nostrils of healthy horses originating from the regions highlighted on the map. * = The black line indicates approximately 3700 km of distance. Created in BioRender. Saraiva, M. (2025) https://BioRender.com/plog7ww.

**Table 1 antibiotics-14-00693-t001:** Cohen’s kappa agreement among *nuc*, *femA*, and *coa* genes, as well as these genes and tube coagulase.

Patterns	Values of Kappa	*p*-Values	Confidence Interval
Coagulase-*femA*	0.295	0.172	sup: 0.719
			inf: −0.128
Coagulase-*nuc*	0.295	0.172	sup: 0.719
			inf: −0.128
*nuc-femA*	0.213	0.33	sup: 0.64
			inf: −0.215
*nuc-coa*	0.025	0.882	sup: 0.361
			inf: −0.31
*nuc-femA-coa*	−0.048	-	sup: 0.199
			inf: −0.295
Coagulase-*coa*	−0.05	-	sup: 0.255
			inf: −0.355
*femA-coa*	−0.152	-	sup: 0.184
			inf: −0.487

Legend: Sup = head limit; inf = lower limit; coagulase = tube-coagulase.

**Table 2 antibiotics-14-00693-t002:** Frequency patterns of isolated forms from *nuc*, *femA*, and *coa* genes.

Patterns of *S. aureus* Markers (Nº)	%	*nuc*	*femA*	*coa*
1 (7)	33.34	+	+	−
2 (6)	28.57	+	+	+
3 (3)	14.29	−	+	−
4 (2)	9.52	+	−	−
5 (2)	9.52	−	−	+
6 (1)	4.76	+	−	+
TOTAL (21)	100			

Legend: (+) = Positive for specific primer; (−) = negative for specific primer.

**Table 3 antibiotics-14-00693-t003:** Resistance and gene patterns of 21 *S. aureus,* as well as origins.

Isolated	Origin	Resistance Patterns	Gene Patterns	Resistance Genes
1	A(IV)	Pan-Susceptible	*Nuc*	-
2	A(V)	CloAziVan	*nuc-femA*	-
3	A(II)	OxaPenCloCfoAmpAmcVan	*Coa*	*blaZ*
4	A(VI)	PenCloAmpTetCliVan	*Coa*	-
5	A(II)	Pan-Susceptible	*nuc-femA*	-
6	A(I)	Pan-Susceptible	*nuc-femA*	-
7	A(I)	Pan-Susceptible	*nuc-femA*	-
8	A(I)	Van	*femA*	-
9	A(I)	Pan-Susceptible	*femA*	-
10	A(VII)	CliEno	*nuc-coa*	-
11	B	PenAmpTetCliVan	*nuc-femA-coa*	-
12	B	CliAmp	*nuc-femA-coa*	-
13	B	PenCloAmpTet	*Nuc*	-
14	B	OxaPenCfoAmpCliGenAzi	*nuc-femA*	*blaZ*
15	B	TetVan	*nuc-femA-coa*	-
16	B	PenCfoAmpTetCliVan	*nuc-femA*	-
17	B	OxaPenCfoAmpTetVan	*nuc-femA-coa*	*blaZ*
18	B	OxaPenCfoAmpCli	*nuc-femA*	*blaZ*
19	B	Pan-Susceptible	*femA*	-
20	B	OxaPenCfoAmpTetCliGen	*nuc-femA-coa*	*blaZ*
21	B	Tet	*nuc-femA-coa*	-

Legend: Amp = ampicillin; Cfo = cefoxitin; Pen = penicillin G; Clo = Chloramphenicol; Gen = gentamicin; Oxa = oxacillin; Cli = clindamycin; Tet = tetracycline; Amc = amoxicillin/clavulanate; Azi = azithromycin; Van = vancomycin; Eno = enrofloxacin.

## Data Availability

The datasets used in this study are available upon request from the corresponding authors.

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
