# Peer review of "Antimicrobial Resistance Patterns of Staphylococcus aureus Cultured from the Healthy Horses’ Nostrils Sampled in Distant Regions of Brazil"

_antibiotics, 2025, doi:10.3390/antibiotics14070693_

Round 1
Reviewer 1 Report
Comments and Suggestions for Authors
-
Please revise the title to follow the title case format, where each significant word starts with a capital letter in accordance with Antibiotics journal guidelines.
-
In the Introduction, the authors should include more data and relevant previous studies on antibiotic resistance in horses or related animal species.
- Line 100: The authors should improve the clarity of the sentence describing antibiotic resistance rates. Additionally, they should specify the total number used to calculate the percentage for each antibiotic.
- The data currently placed in the supplementary section should be moved to the Methods section of the manuscript.
- The authors should add labels (A and B) to the panels in Figure 2.
Author Response
Comments 1: Please revise the title to follow the title case format, where each significant word starts with a capital letter in accordance with Antibiotics journal guidelines.
Authors' reply: Thank you for your comment and for giving us the opportunity to improve our manuscript. Both the title and the entire manuscript are now in accordance with journal guidelines. We invited the Reviewer to find the new version of the manuscript with improvements highlighted in red.
Comments 2: In the Introduction, the authors should include more data and relevant previous studies on antibiotic resistance in horses or related animal species.
Authors' reply: Done. More information has been added in the introduction section (Lines: 65-66; 76-81), as well as four new references (doi.org/10.1016/j.jevs.2023.104498; doi.org/10.1111/evj.13437; doi.org/10.3390/antibiotics12020242; doi.org/10.2460/javma.24.11.0732).
Comments 3: Line 100: The authors should improve the clarity of the sentence describing antibiotic resistance rates. Additionally, they should specify the total number used to calculate the percentage for each antibiotic.
Authors' reply: Done (Lines 110-117).
Comments 4: The data currently placed in the supplementary section should be moved to the Methods section of the manuscript.
Authors' reply: Thank you for the Reviewer's comment. Although we perfectly understand the Reviewer's point, all substantial data to comprehend the carrying out and results of the manuscript have been included in the Materials and Methods section. We understand that the primer sequences and Kapp-Cohen index criteria could be accessed by the reader as supplementary material, without the necessity of adding two more tables to the main manuscript. Therefore, we cordially ask that the Reviewer evaluate the new manuscript version from this perspective.
Comments 5: The authors should add labels (A and B) to the panels in Figure 2.
Authors' reply: Done (Line 250).
Reviewer 2 Report
Comments and Suggestions for Authors
- Please, correct citation style of references in the manuscript. The references must be cited in text with number.
- It will be better to clearly explain the main goal and outcomes of these research in abstract and introduction part.
- I suggest uploading approvement document by Ethical Committee.
Author Response
Comments 1: Please, correct citation style of references in the manuscript. The references must be cited in text with number.
Authors' reply: Thank you for your comment and for giving us the opportunity to improve our manuscript. The citation style, as well as the entire manuscript, is now in accordance with journal guidelines. We invited the Reviewer to find the new version of the manuscript with improvements highlighted in red.
Comments 2: It will be better to clearly explain the main goal and outcomes of these research in abstract and introduction part.
Authors' reply: Done (Lines: 25-30; 41-42; 87-91).
Comments 3: I suggest uploading approvement document by Ethical Committee.
Authors' reply: Thank you for your comment. We perfectly understand the importance of the Ethical Committee Document. However, in this specific research, the bacteria used were previously stored in the bacteriotheca. Therefore, a formal process of the Ethical Committee was not necessary for this study, according to the College for Agricultural Sciences, Federal University of Paraíba (CCA/UFPB) rules.
Reviewer 3 Report
Comments and Suggestions for Authors
The manuscript entitled "Antimicrobial Resistance Patterns of S. aureus Cultured from Nostrils of Healthy Horses from Distant Geographical Regions in Brazil." investigates the antimicrobial resistance profiles and genotypic diversity of Staphylococcus aureus isolated from the nasal cavities of healthy horses in two distant Brazilian regions. The study addresses significant One Health concerns regarding MDR and BORSA strains in animal reservoirs. The methodology is generally sound, and the findings contribute to understanding equine-associated S. aureus epidemiology. However, some aspects require clarification and refinement to enhance the clarity and scientific rigor of the work.
Author affiliations list " 5 Center for Computational Intelligence to Predict Health and Environmental Risks (CIPHER), The University of North Carolina at Charlotte, Charlotte, USA"; however, no specific author appears to be associated with this institution. Please clarify.
Please consider slightly rephrasing the title for clarity. For example, “from the nostrils of healthy horses sampled in distant regions of Brazil” may sound smoother.
In the abstract, the expression "highlighting significant genetic diversity" is appropriate, but please consider briefly commenting on the implications of this diversity for surveillance or animal-human transmission.
The rationale behind focusing on horses should be more clearly emphasized. For instance, why horses were selected over other livestock species could be briefly explained.
Was convenience sampling applied, or were there specific criteria for selecting horses? Please specify.
Please indicate the reference for interpreting the antimicrobial susceptibility test results, such as CLSI guidelines or another recognized standard.
If possible, please elaborate on whether the lack of clustering by geographic region was expected or surprising and briefly comment on what this could indicate epidemiologically.
The discussion thoroughly interprets the findings; however, please consider softening speculative statements (e.g., “this condition implies the treatment failure”) by indicating they are potential rather than definitive outcomes.
When mentioning “pleomorphism in mec-operon,” please clarify whether sequencing was performed to support this claim or if it is inferred from the literature.
Author Response
Comment: The manuscript entitled "Antimicrobial Resistance Patterns of S. aureus Cultured from Nostrils of Healthy Horses from Distant Geographical Regions in Brazil." investigates the antimicrobial resistance profiles and genotypic diversity of Staphylococcus aureus isolated from the nasal cavities of healthy horses in two distant Brazilian regions. The study addresses significant One Health concerns regarding MDR and BORSA strains in animal reservoirs. The methodology is generally sound, and the findings contribute to understanding equine-associated S. aureus epidemiology. However, some aspects require clarification and refinement to enhance the clarity and scientific rigor of the work.
Authors' reply: Thank you for your comment and the opportunity to improve our manuscript. We invited the Reviewer to find the new version of the manuscript with improvements highlighted in red.
Comment: Author affiliations list " 5 Center for Computational Intelligence to Predict Health and Environmental Risks (CIPHER), The University of North Carolina at Charlotte, Charlotte, USA"; however, no specific author appears to be associated with this institution. Please clarify.
Authors' reply: Done (Line 6).
Comment: Please consider slightly rephrasing the title for clarity. For example, “from the nostrils of healthy horses sampled in distant regions of Brazil” may sound smoother.
Authors' reply: Thank you for the suggestion. We reformulate the title, according Reviewer's suggestion (Lines 2-4).
Comment: In the abstract, the expression "highlighting significant genetic diversity" is appropriate, but please consider briefly commenting on the implications of this diversity for surveillance or animal-human transmission.
Authors' reply: Thank you for your suggestion. A brief comment has been added (Lines 41-42).
Comment: The rationale behind focusing on horses should be more clearly emphasized. For instance, why horses were selected over other livestock species could be briefly explained.
Authors' reply: Thank you for the comment. We performed improvements in the Introduction section (Lines: 65-66; 76-83; 87-91).
Comment: Was convenience sampling applied, or were there specific criteria for selecting horses? Please specify.
Authors' reply: Thank you for the comment. All the equines were randomly sampled. This information has been added in the Materials and Methods section (Line 246).
Comment: Please indicate the reference for interpreting the antimicrobial susceptibility test results, such as CLSI guidelines or another recognized standard.
Authors' reply: Done (Lines 284-285).
Comment: If possible, please elaborate on whether the lack of clustering by geographic region was expected or surprising and briefly comment on what this could indicate epidemiologically.
Authors' reply: Done (Lines 231-234; 240-242).
Comment: The discussion thoroughly interprets the findings; however, please consider softening speculative statements (e.g., “this condition implies the treatment failure”) by indicating they are potential rather than definitive outcomes.
Authors' reply: Thank you for the suggestion. We invite the Reviewer to find the improved version of the Discussion section.
Comment: When mentioning “pleomorphism in mec-operon,” please clarify whether sequencing was performed to support this claim or if it is inferred from the literature.
Authors' reply: Thank you for the comment. It was inferred from the literature. We modified the sentence (Lines 187-190).
Reviewer 4 Report
Comments and Suggestions for Authors
Line 49, please write: Multi-Drug Resistance (MDR) strains
Line 86, please write: coagulase-negative Staphylococci (CoNS)
Author Response
Authors' comment: We deeply appreciated the care and contribution to manuscript improvements and the paper publication. We invited the Reviewer to find the new version of the manuscript with improvements highlighted in red.
Comment: Line 49, please write: Multi-Drug Resistance (MDR) strains.
Authors' reply: Done (53-54).
Comment: Line 86, please write: coagulase-negative Staphylococci (CoNS).
Authors' reply: Done (96).
Round 2
Reviewer 1 Report
Comments and Suggestions for Authors
The authors revised the comments point by point.